

# Kelpie: generating full-length 'amplicons' from whole-metagenome datasets

Paul Greenfield[1,2,3], Nai Tran-Dinh[1] and David Midgley[1]

[1] Commonwealth Scientific and Industrial Research Organisation, North Ryde, NSW, Australia
[2] School of Biological Sciences, Macquarie University, Australia
[3] School of Information Technologies, University of Sydney, Australia

## ABSTRACT

**Introduction**. Whole-metagenome sequencing can be a rich source of information about the structure and function of entire metagenomic communities, but getting accurate and reliable results from these datasets can be challenging. Analysis of these datasets is founded on the mapping of sequencing reads onto known genomic regions from known organisms, but short reads will often map equally well to multiple regions, and to multiple reference organisms. Assembling metagenomic datasets prior to mapping can generate much longer and more precisely mappable sequences but the presence of closely related organisms and highly conserved regions makes metagenomic assembly challenging, and some regions of particular interest can assemble poorly. One solution to these problems is to use specialised tools, such as Kelpie, that can accurately extract and assemble full-length sequences for defined genomic regions from whole-metagenome datasets.

**Methods**. Kelpie is a kMer-based tool that generates full-length amplicon-like sequences from whole-metagenome datasets. It takes a pair of primer sequences and a set of metagenomic reads, and uses a combination of kMer filtering, error correction and assembly techniques to construct sets of full-length inter-primer sequences.

**Results**. The effectiveness of Kelpie is demonstrated here through the extraction and assembly of full-length ribosomal marker gene regions, as this allows comparisons with conventional amplicon sequencing and published metagenomic benchmarks. The results show that the Kelpie-generated sequences and community profiles closely match those produced by amplicon sequencing, down to low abundance levels, and running Kelpie on the synthetic CAMI metagenomic benchmarking datasets shows similar high levels of both precision and recall.

**Conclusions**. Kelpie can be thought of as being somewhat like an *in-silico* PCR tool, taking a primer pair and producing the resulting 'amplicons' from a whole-metagenome dataset. Marker regions from the 16S rRNA gene were used here as an example because this allowed the overall accuracy of Kelpie to be evaluated through comparisons with other datasets, approaches and benchmarks. Kelpie is not limited to this application though, and can be used to extract and assemble any genomic region present in a whole metagenome dataset, as long as it is bound by a pairs of highly conserved primer sequences.

Corresponding author
Paul Greenfield,
paul.greenfield@csiro.au

## INTRODUCTION

Kelpie can be thought of as an *in silico* PCR program. It takes a pair of primer sequences and a whole metagenome sequencing (WGS) dataset, and generates a corresponding set of inter-primer amplicon-like sequences. It does this using iterative kMer filtering, error correction, incremental assembly and recursive tree exploration. The results section of this paper primarily discusses using Kelpie to determine the composition of a metagenomic community, although this is just one possible application, and was chosen simply because of the availability of suitable datasets and benchmarks, including sets of 'correct' results for effectiveness comparisons.

Whole-metagenome sequencing datasets can be a rich resource for investigating both the structure of a metagenomic community and the functional capabilities of its members, but reliably and accurately extracting such information from large volumes of sequencing data can be challenging. These challenges arise from the nature of the sequencing data itself, the presence of ubiquitous and highly conserved genomic regions and the possible presence of related organisms within the community. Whole genome sequencing (WGS) metagenomic data is typically generated using a platform such as Illumina HiSeq or NovaSeq. These systems produce very large volumes of short (100–150 bp) reads at a low cost per read, but their short length makes them less distinct than longer reads would be, and so more difficult to map unambiguously to known reference sequences for the purposes of classification and annotation. Assembling the metagenomes can generate much longer and more distinctive sequences, and these can be used to more reliably determine the presence of particular organisms or genes, but metagenomic assembly is itself challenging in the presence of conserved regions and related organisms (*Treangen & Salzberg, 2012*; *Wang et al., 2015*).

The challenges involved in accurately interpreting short metagenomic datasets are well illustrated by the task of determining the structure of a community from a WGS metagenomic dataset. Approaches based on the direct mapping of reads face issues arising from the indistinctiveness of short reads, while assembly-based approaches run into problems caused by conserved regions and related organisms.

Metagenomic community profiling is commonly done using low cost targeted amplicon sequencing, rather than through the analysis of whole genome sequencing datasets. A chosen variable region of a marker gene is extracted and amplified from metagenomic DNA using multiple rounds of PCR, with a pair of primers that match highly-conserved sequences on either side of the region. These primers can be chosen to cover regions that are sufficiently long and variable to allow closely related organisms to be distinguished from each other. The resulting 'amplicons' are then sequenced, and the reads are then processed and classified in some way to determine which organisms are present in each sample and to give some idea of their abundances. Given a well-chosen variable region and suitable primers, and a good quality reference gene database, this approach can accurately identify the organisms present in a community, perhaps down to species level. Amplicon sequencing is also well supported through tools, pipelines and reference datasets. The book (*Taberlet et al., 2018*) gives an overview of amplicon sequencing in the context of environmental DNA studies, and discusses both its strengths and weaknesses.

Whole genome sequencing can answer more questions than amplicon sequencing as its reads are derived from the entire genomes of the community members, rather than just being constrained to a small region of a single chosen marker gene. Community structure can be determined from WGS datasets through the use of either marker gene or whole genome comparisons; and both of these approaches can be based on either the WGS reads directly or on the contigs coming from a metagenomic assembly of these reads. The report from the Critical Assessment of Metagenome Interpretation (CAMI) benchmarking project (*Sczyrba et al., 2017*) discusses these different approaches in some detail, and compares the effectiveness of a number of published WGS community profiling tools on a set of synthetic datasets. The paper from *Lindgreen, Adair & Gardner (2016)* also presents an overview of available metagenome analysis tools.

The first step in those workflows that are based on mapping WGS reads to marker genes is to search the datasets for just those reads that appear to be derived from the chosen marker gene. This search can be done with tools based on Hidden Markov Models (HMM), a fast kMer filter (*Greenfield, 2018a*), or even BLASTing all the WGS reads against a reference set. The result will be a small set of filtered reads, around 0.1% of the initial WGS read set for the bacterial 16S rRNA gene. These reads can then be classified in conventional ways, such as through the use of statistical classifiers, such as RDP (*Wang et al., 2007*), or by matching them to a reference database. The short length and random placement of the WGS reads reduces the effectiveness of this approach, as some of the selected reads will come from conserved regions of the marker gene or from related organisms, and may resolve only to a higher taxonomic levels, such as Class or Order. The EBI Metagenomics Portal (*Mitchell et al., 2018*) uses this reads-based approach, first filtering the WGS reads using Infernal (*Nawrocki & Eddy, 2013*) in HMM-only mode against a library of ribosomal RNA models from Rfam (*Nawrocki et al., 2015*), and then classifying these selected reads with MAPseq (*Matias Rodrigues et al., 2017*) against a SILVA SSU/LSU reference set (*Quast et al., 2013*).

Given the resolution limitations inherent with short WGS reads, an appealing alternative approach is to first assemble the WGS metagenomic datasets, and then search for the wanted marker genes in the resulting contigs. The gene sequences that are found can then be classified in any of the usual ways. Metagenomic assembly can potentially produce complete marker gene sequences, resulting in much improved classification resolution, perhaps down to the level of species or strain. In practice though, the presence of highly conserved regions in these marker genes makes their accurate assembly very challenging, and the resulting genomic regions are often incomplete or split into multiple small contigs, with consequent impacts on the accuracy of the resulting community profile. The limitations of this assembly-based approach are discussed further in the Results section.

Gene-targeted assembly can also be used to extract longer and more classifiable sequences from metagenomic WGS datasets. Tools such as EMIRGE (*Miller et al., 2011*) basically align WGS reads to a set of reference sequences, and then make adjustments to these alignments to produce complete target region sequences. Xander (*Wang et al., 2015*) is another targeted assembler that works by first building a de Bruijn graph from the WGS reads and then searching for regions that match HMMs generated from reference sequences for the target

genes. Both these approaches effectively work by aligning WGS reads to sets of reference sequences, and rely on genes present in the community being close enough to ones found in the reference sets to get sufficiently unambiguous alignments. Kelpie has no such dependency on reference sequences and, like PCR, just takes a pair of primer sequences and returns whatever was found between them, regardless of its similarity to known genes or conformity to a model trained on such genes.

Taxonomic profiling can also be based on whole genomes rather than just marker genes. WGS reads or assembled contigs can be matched against whole genome reference sets, and the results used to generate community profiles. The appeal of this approach is that it may better separate closely related organisms, especially if these differ in their functional capability through horizontal gene transfer or the acquisition of plasmids. Megan (*Huson et al., 2011*), Kraken (*Wood & Salzberg, 2014*) and MG-RAST (*Glass et al., 2010*) are examples of tools and pipelines that are based on whole genome profiling. In practice, the effectiveness of this approach is limited by the restricted taxonomic coverage of the available reference sets, especially for environmental studies where novel organisms are commonplace. Some of the tools considered by the CAMI study are based on whole genome profiling and will be discussed further in the Results section.

WGS taxonomic profiling with Kelpie starts by extracting and assembling sets of full-length amplicon-like marker gene sequences from a set of filtered WGS reads and pair of primer sequences. These 'amplicon' sequences can then be run through conventional amplicon pipelines or other such tools to generate taxonomic profiles and further results of interest. This approach can result in improved resolution compared to direct read mapping as the assembled sequences can be considerably longer than a single WGS read, come from known regions within a chosen marker gene, and the primers can be chosen to cover informative variable regions. Kelpie generates sets of extended reads rather than traditional contigs, with each of these amplicon-like sequences being seeded from a single WGS read that contained the specified forward primer that was then extended until a reverse primer sequence was reached. Generating sets of extended reads in this way makes Kelpie compatible with conventional amplicon-based pipelines, and also tends to preserve some strain variation, as discussed in the Results. In practice, taxonomic profiles generated using Kelpie are highly accurate, and give comparable results to PCR-based amplicon sequencing, down to the point where there is insufficient depth of coverage in the WGS dataset to fully cover the chosen marker gene regions.

The results presented later are all based on this application as it allowed the effectiveness and accuracy of Kelpie to be assessed through comparison against alternative techniques and published metagenomic benchmarks. Kelpie is not limited to this application though, and can be used to extract and assemble any genomic region present in a WGS dataset that is bounded by pairs of highly conserved primer sequences.

## METHODS & MATERIALS

### Algorithm description

Kelpie is founded on the distinctiveness properties of medium to large kMers (>∼20 bp). Once 'k' is big enough, the space of possible kMers becomes so large that instances of

kMer sharing between organisms almost always signify shared genes or domains, either through relatedness or gene transfer (*Greenfield & Roehm, 2013*). One useful application of this distinctiveness property is that, given a long enough kMer, it is frequently possible to correctly predict the kMer that follows it in the genome from which it was derived, even with metagenomic datasets. These predictions are done by simply generating each of the 4 possible following kMers, produced by concatenating the rightmost (k-1) bases of the current kMer with each of 'A', 'C', 'G', and 'T' in turn, and checking the presence of each these variants in the set of distinct kMers constructed from the entire WGS dataset. For Kelpie running on the three coal seam metagenomes discussed in the Results section, this concatenate-and-check technique found that there was only a single viable 'next' kMer 99.6% to 99.9% of the time when attempting to extend an under-construction amplicon by a single base at a time. This technique of generating long sequences through unambiguous extension is also at the core of kMer-based error correction algorithms, such as Blue (*Greenfield et al., 2014*), and is also used in the Inchworm phase of the Trinity RNA transcript assembler (*Grabherr et al., 2011*).

In the current release of Kelpie, the starting point is a filtered subset of an entire WGS dataset that just contains reads derived from the genomic region of interest, such as the 16S or 18S rRNA gene. For a bacterial metagenomic dataset, this initial filtering will typically reduce the data volumes to be processed by Kelpie by around 99.9%, making it feasible to keep the filtered reads in memory for much improved performance. This filtering does not have to be exact, and including some non-target reads will have little effect as they will be discarded by the more targeted filtering performed in the first stage of Kelpie. This kind of filtering is also used in other reads-based WGS metagenomic taxonomic profiling pipelines, such as the EBI Metagenomics Portal (*Mitchell et al., 2018*). It is also helpful if the filtered reads are quality-trimmed before being processed by Kelpie, as this will reduce the number of erroneous kMers that have to be considered. This can be done within filtering tool if it has an appropriate option, or separately with a tool such as Trimmomatic (*Bolger, Lohse & Usadel, 2014*).

Kelpie processes a filtered WGS dataset in three distinct phases:

- Extracting just those reads that cover the defined inter-primer region;
- Building kMer tables from these inter-primer region reads;
- Extending any reads found to contain a forward primer sequence.

The first phase goes through the initial filtered reads and uses an iteratively constructed inter-primer kMer filter to select just those that cover the specified inter-primer region. The first step in building this filter is to find all the reads that contain a 'starting' primer sequence (either the specified forward primer or the reverse complement of the reverse primer). The kMers (actually 32-mers) following the primers in these starting reads are added to an initial inter-primer filter (a kMer hash set). The remaining reads are then scanned again, looking for any that start with a kMer found in this inter-primer filter set. The kMers from these newly selected reads are then added to the inter-primer filter, and the remaining reads are scanned once again. This process continues until the entire inter-primer region has been covered and the reads being selected contain a 'finishing' primer. The initial set

of filtered reads are then passed over this inter-primer filter once again, and just those reads that start with kMers found in the filter set are retained. Any reads that contain either 'starting' or 'finishing' primer sequences are trimmed appropriately. The result is a set of trimmed reads that cover the primer-defined region. Any reads starting with the forward primer or ending with its reverse complement are marked as 'starting', and these reads are the ones that will be extended in the final phase.

The next phase turns these 'inter-primer' reads into the collection of kMer hash tables for the extension phase. There is no ideal length for these extending kMers. Shorter kMers, such as 32-mers are more plentiful but are also more likely to be shared between different organisms; while longer kMers (such as 80-mers) are more distinctive and less likely to be ambiguously shared, but fewer of them can be derived from each read, especially from the trimmed reads at the start and end of the primer-defined region. The solution adopted is to have multiple kMer tables, with 'k' ranging from 32 to almost the full read length (in steps of 8 bp). For 100 bp reads, this results in nine such kMer hash tables. The reads are first tiled for 32-mers and the resulting kMer table is built and then 'denoised' by removing dubious kMers, such as those found only once, or those that are rare and appear to be error variants of abundant kMers. The reads are tiled repeatedly for progressively longer kMers to construct the remaining kMer tables.

The final phase then takes each of the marked 'starting' reads in turn and calls ExtendRead to try to unambiguously extend it, one base at a time, until the extended read finishes with a 'finishing' primer. At every iteration of the extension loop, the read is effectively further extended by adding each of the possible bases, A, C, G and T to its end. Each of these possible extended reads is then checked against the full collection of kMer tables to ensure that it is fully supported by the WGS reads. If multiple extensions prove to be viable, each of them is then tested to see if extending it further would eventually reach a terminal primer, and so result in a complete amplicon. This extension check is done by recursively calling ExtendRead on each of the viable extensions. Figure 1 contains a high-level pseudocode description of this ExtendRead method and its iterative and recursive extension loop.

The presence of strains and closely related organisms within metagenomic communities means that sometimes there will be multiple viable extended reads, all of which have both good kMer support from the WGS data and end in a terminal primer. In this case, Kelpie breaks the tie by randomly choosing one of the reads, in proportion to the repetition depth of the 32-mers at the end of the read being iteratively extended. For example, if both 'A' and 'T' could viably be added to the end of the extending read, and the 32-mer xxxxxxxxA was found 90 times in the WGS reads, and the xxxxxxxxT 32-mer was found 10 times, the Kelpie will choose the 'A'-extended read 90% of the time. Choosing in proportion in this way ensures that less common strain variants are properly represented in the final set of 'amplicons', and not just subsumed by more common variants.

Once all the 'starting' reads have been extended, Kelpie drops any of these extended reads that did not reach a terminating primer, trims the primer sequences from the remaining full-length amplicons and writes them out as a FASTA file.

The depth-first recursive exploration of possible read extensions is computationally expensive, but, in practice this path is rarely taken. Almost all decisions about what base to

```
method ExtendRead(read, out extendedRead, out tpReached)
{
   extendedRead = read;
   while (extending)
   {
      // does the extended read end with a terminal primer?
      if (extendedRead.EndsWith(terminalPrimer))
         return extendedRead & 'true';

      // check whether the 4 possible extended reads all have support from the
      // WGS reads via the kMerTables
      foreach (kMerLength in kMerTableLengths)
      {
         generate all 4 possible 'next' (kMerLength) kMers at end of the read;
         lookup their counts in (kMerLength) kMerTable and check if viable;
         save 32-mer counts for later tie breaking;
         // if none of the extensions are viable, abandon this read extension
         if (viableAlternatives == 0)
            return extendedRead and 'false';
         // if just one extension is viable, so stop checking longer kMers
         if (viableAlternatives == 1)
            break;
      }

      // only one of the possible extensions is viable
      if (viableAlternatives == 1)
      {
         // add the viable base to the read and continue extending
         extendedRead += viableBase;
         continue;
      }

      // multiple viable extensions…
      // recursively explore each of them and see how far downstream it can get
      generate viableExtendedReads[] from all viable extensions;
      foreach (read[i] in viableExtendedReads[])
         // recursively explore the consequences of this extension
         tpReached[i] = ExtendRead(read[i], out read[i]);

      count number of extensions that reached terminal primer (tpCount);
      // none of the extensions get to the end…
      if (tpCount == 0)
         return longest of extended reads and 'false';
      // just one extension could get to terminal primer, so all done
      if (tpCount == 1)
         return only winning read (now fully extended) and 'true';

      // tie: multiple extensions can reach a terminal primer
      randomly choose a winner in proportion of saved 32-mer counts;
      return winning (now fully extended) read and 'true';

   } // while (extending)
}
```

**Figure 1  Pseudocode for Kelpie extension phase.**

choose to further extend an extending read are taken just by looking at the kMer tables, and almost all of these decisions only have to check the initial 32-mer table. Table 1 presents some statistics on these decisions from the three coal seam metagenomes discussed in the 'Results' section. The initial 32-mer kMer check came up with a single unambiguous 'next' base in 96.0% to 98.7% of the time, and checking the remaining kMer tables came up with a single choice 98.2% to 100.0% of the time. It was only necessary to explore the tree of possible extensions 0.0% to 1.2% of the time. One necessary limitation with Kelpie, in common with all other assemblers, is that all the distinct gene regions being extended/assembled have to be completely covered by reads from the WGS dataset. The

**Table 1  Read extension decision statistics for three CSM datasets.**

|  | W1 | W2 | W3 | W1 | W2 | W3 |
|---|---|---|---|---|---|---|
| read extension checks | 20517124 | 3562698 | 8402118 |  |  |  |
| single choice at $k = 32$ | 19700062 | 3518077 | 8222559 | 96.0% | 98.7% | 97.9% |
| single choice at $k = 40$ | 66318 | 15118 | 77026 | 0.3% | 0.4% | 0.9% |
| single choice at $k = 48$ | 72763 | 12650 | 11615 | 0.4% | 0.4% | 0.1% |
| single choice at $k = 56$ | 17655 | 715 | 10485 | 0.1% | 0.0% | 0.1% |
| single choice at $k = 64$ | 34950 | 1488 | 4043 | 0.2% | 0.0% | 0.0% |
| single choice at $k = 72$ | 19292 | 11136 | 21155 | 0.1% | 0.3% | 0.3% |
| single choice at $k = 80$ | 18425 | 1128 | 7107 | 0.1% | 0.0% | 0.1% |
| single choice at $k = 88$ | 117494 | 0 | 7540 | 0.6% | 0.0% | 0.1% |
| single choice at $k = 96$ | 97699 | 1273 | 3204 | 0.5% | 0.0% | 0.0% |
| single kMer choice | 20144658 | 3561585 | 8364734 | 98.2% | 100.0% | 99.6% |
| looked downstream | 238307 | 962 | 28396 | 1.2% | 0.0% | 0.3% |
| single good downstream | 134009 | 46 | 7436 | 0.7% | 0.0% | 0.1% |
| chose in proportion by depth | 104293 | 909 | 19487 | 0.5% | 0.0% | 0.2% |
| chose longest downstream | 5 | 7 | 1473 | 0.0% | 0.0% | 0.0% |
| # of starting reads | 19750 | 15876 | 23324 |  |  |  |
| # of reads abandoned | 145 | 28 | 79 | 0.7% | 0.2% | 0.3% |
| # of fully extended reads | 19605 | 15848 | 23245 | 99.3% | 99.8% | 99.7% |

impact of this requirement is that rarer organisms in the community will not be represented in the set of extended reads if their coverage is incomplete.

All of the results discussed in the next section were obtained from running Kelpie (V1.0.3) on a Dell Latitude E7470 laptop with a 2.4 GHz Intel i5-6300U processor (2 cores, 4 threads) and 16GB of RAM. The three coal seam microbiome datasets took 80.0 s, 33.7 s and 73.1 s to process, after the preliminary 16S rRNA filtering had been done. The CAMI Low and Medium datasets took 27.3 s and 33.1 s, respectively, after filtering. Kelpie is written in C# and can be run under Windows, OSX and Linux. Kelpie code is open source and available for download from GitHub (*Greenfield, 2018b*). It is made available under an MIT licence.

Kelpie is a command-line program and is usually run as follows:

```
Kelpie -f forwardPrimer -r reversePrimer readsToFilterFNP extendedReadsFN
```
where `forwardPrimer` is the forward primer sequence
`reversePrimer` is the reverse primer sequence
`readsToFilterFNP` is a list of reads file names or a file name pattern
`extendedReadsFN` is the file name for the extended reads
For example: `Kelpie -f GTGYCAGCMGCCGCGGTAA -r GGACTACNVGGGTWTCTAAT`
`                    W1_?_16S_20_fz_25.fa W1_16S_v4.fa`

Other options are available for use in very unusual cases, and these are described in the documentation provided as part of the Kelpie package.

## TESTING & EVALUATION

The effectiveness and accuracy of Kelpie was evaluated using its application to the task of determining the structure of a metagenomic community. This application was chosen because the Kelpie-generated results could be compared both against independently defined 'truths' and results from alternative tools and techniques. The results generated by Kelpie were compared to:

- a profile produced from the EBI Metagenomics pipeline (*Mitchell et al., 2018*).
- real amplicon data from a coal seam metagenome project.
- two of the synthetic metagenomic datasets generated by the CAMI project (*Sczyrba et al., 2017*).

The amplicon PCR used the 16S rRNA V4 primers defined by the Earth Microbiome Project (*Thompson et al., 2017*). These primers were GTGYCAGCMGCCGCGGTAA (forward) and GGACTACNVGGGTWTCTAAT (reverse). These two primers were used in the corresponding Kelpie tests so that the results from real amplicon sequencing and Kelpie could be meaningfully compared, and the same primers were also used for the EBI and CAMI-based tests.

The Kelpie numbers presented in the results below are counts of fully-extended reads. As discussed earlier, Kelpie works by finding reads that include a starting primer sequence and extending each one of these, one base at a time, until a terminating primer is reached. The result is a set of full-length extended reads, each one of which has its origins in a single WGS read.

### EBI metagenomic pipeline

The starting point for Kelpie is currently, for performance reasons, a set of filtered reads that cover the target region, and probably the entire target gene that surrounds it. The EBI Metagenomics pipeline includes such a 16S rRNA filtering step, and the selected reads are directly classified to generate a taxonomic profile for the submitted dataset. As part of the testing of Kelpie, an EBI Illumina WGS submission (MGYS00000465/ERP008951) was chosen at random. The 16S rRNA HMM-filtered reads were downloaded from EBI, assembled with Kelpie using the above v4 primers, and run through the GHAP amplicon pipeline (*Greenfield, 2017*). This test both demonstrated that Kelpie was compatible with 16S rRNA reads selected using conventional HMM-based tools, and also allowed a comparison between the profiles produced from the Kelpie-assembled reads and the results generated by the EBI pipeline.

### Coal seam metagenomes

Three coal seam metagenome (CSM) samples, called here W1, W2 and W3, were produced as part of an industry-funded study of microbial life in Queensland coal fields. Two of the samples came from the Surat Basin and the other from the Bowen Basin. The DNA extracted from each of these three samples was split, with one part being sent off for WGS sequencing (paired-end 100 bp Illumina HiSeq), and the other amplified using the EMP V4 PCR primers and then sequenced (paired-end 300 bp Illumina MiSeq). The amplicon

and WGS datasets for all three samples are available for download at Coal Seam Formation Water Community Profiles (*Greenfield, 2018c*). These three samples are also included in the Coal Seam Microbiome reference set (*Vick et al., 2018*) as Surat 3 (W1), Bowen 3 (W2) and Surat 1 (W3).

The amplicon data was quality-trimmed and then pair-merged using the USearch *fastq_mergepairs* function (*Edgar, 2010*) to produce full-length (∼250 bp) amplicon sequences (with primers trimmed).

The WGS data was filtered for just the reads that covered the 16S rRNA gene by a simple kMer filter (*Greenfield, 2018a*) that kept only those reads that had sufficient kMer matches onto genes included in a 16S rRNA reference set. This reduced the size of the data files to be processed by Kelpie by about 99.8% (filtering the W3 WGS dataset looked at 322,127,528 reads and kept just 597,668 of them). These putative 16S reads were then processed with Kelpie using the same EMP V4 primers to produce a set of full-length amplicon-like sequences.

The Kelpie-generated and the real amplicon datasets were then run through a conventional amplicon pipeline (*Greenfield, 2017*) based on USearch and RDP (*Wang et al., 2007*) tools, both separately and together. The results were assessed to see if similar community profiles were generated from both sets of reads, and if the actual sequences generated by PCR and Kelpie were identical.

A more conventional way of extracting inter-primer regions from WGS metagenomic data would be to first assemble the full dataset using a metagenomic assembler, and then search for and extract just the targeted genomic regions from the resulting contigs. This approach was evaluated by first assembling the full WGS dataset for each of the three samples with metaSPAdes (*Nurk et al., 2017*), and searching within the resulting contigs for regions bounded by the specified forward and reverse primers. metaSPAdes was also used to assemble just the filtered 16S rRNA reads used as input to Kelpie, as it was thought that this could result better assemblies.

## CAMI synthetic benchmarks

One shortcoming with the previous two sets of tests is that they are based on comparisons against results produced using conventional tools and techniques, and these alternative approaches have their own imperfections and quirks. A better approach would be to test Kelpie using a dataset with known 'correct' answers and then calculate accepted performance statistics, such as *recall* and *precision*. The datasets produced for the CAMI Challenge (*Sczyrba et al., 2017*) come close to meeting this requirement, providing sets of synthetic metagenomic reads derived from known organisms, mostly named to the species level. The CAMI paper both defines standard performance metrics, and compares the performance of a wide variety of published tools on these benchmark datasets.

The CAMI challenge datasets consist of synthetically generated reads produced from assembled contigs built from sequence data generated from named cultured organisms. This approach gave the CAMI organisers considerable flexibility, allowing them to produce both Illumina HiSeq-like reads and long mate-pair reads with known error rates, and to simulate the presence of multiple strains of a single starting organism. These reads

were generated only from assembled contigs, not from complete genomes or from the unassembled sequence data, and any flaws or gaps in the assemblies are consequently reflected in the reads used in the study. The ribosome is always challenging to assemble because it is usually present multiple times in each organism in the metagenome, and this high level of replication often results in ribosomal sequences being broken into multiple small contigs. The CAMI synthetic reads were generated only from contigs greater than 1 Kbp in length, and any genes found only in the ignored shorter contigs will be missing from the corresponding WGS datasets.

The first step in testing Kelpie against the low and medium complexity CAMI datasets was to determine which organisms actually had their requisite 16S rRNA regions covered by the generated synthetic reads. This was done by taking the CAMI-provided contig files and extracting all regions bounded by the same primer sequences used in the amplicon study above. These extracted 16S rRNA V4 regions were then matched (using *usearch_global*) against a collection of 16S rRNA RefSeq reference sequences (downloaded from GenBank on 23/July/2017). The result was a set of sequences and accession names for those organisms found to have complete marker gene regions present in the size-filtered contigs. Only these sequences would have been available to the synthetic read generation process, and so be represented in the provided WGS datasets.

These sets of named organisms were then compared to the CAMI-provided 'gold' taxonomic profiles for each of the datasets. There were no matching contig-derived sequences for 7 of the 25 species in the low complexity profile, and for 24 of the 95 species in the medium complexity profile. These missing species were consequently excluded from the performance evaluations as reads derived from their 16S rRNA V4 regions must also be missing from the WGS datasets.

Most of the organisms in the CAMI profiles were named to the Species level, although a few were classified only to higher levels such as Family. The species names from the accessions and the CAMI profile were almost always identical. One of the low complexity organism sequences (*Anaerobranca*) matched at 100% identity to an accession with a different species in the same genus; and four of the organisms in the medium community profile matched different species, again in the same genus. In those cases where the CAMI profiles did not go down to Species, the matched accessions were all taxonomically compatible with the stated classification. The CAMI challenge was also interested in seeing how well the tools under test could separate strains of species present in a community, and so simulated the presence of strain variants for some of organisms in the medium complexity dataset (for 25 of the 95 species). Some of these 'strains' resulted in their own named accessions, and so were included in the Kelpie comparisons.

The CAMI synthetic WGS datasets reflect real metagenomic sequencing datasets in that the organisms in the community are present at different abundance levels, raising the possibility that low abundance organisms may have incomplete coverage of the marker gene region even if was present in the contigs used to generate the WGS reads. As an example, the lowest abundance organism whose 16S rRNA V4 region was found in the contigs was *Nonlabens dokdonensis* at 0.08% abundance. Given the number of 150-mer WGS reads in the dataset, and the size of the *Nonlabens* genome (3,914,632 bp), the estimated depth of

coverage of this organism is 3.2. As marker sequences with incomplete coverage cannot be assembled with Kelpie, a kMer depth of coverage was calculated for each of the extracted marker gene regions.

Kelpie was then run on the both CAMI Low and Medium complexity WGS datasets, using the same the 16S rRNA V4 primers sequences as used in the other two tests. The resulting full-length extended sequences were then matched against the same RefSeq-based 16S reference set, again using *usearch_global*, and the matches were summarised into a set of matched species/strains and counts to create taxonomic profiles. These Kelpie-based profiles were then aligned with the 'gold' and 'found in contigs' profiles to generate combined tables of organisms and numbers of matching reads. As the CAMI datasets were derived from known cultured organisms, the results from testing Kelpie against these datasets can be evaluated using the same *precision/recall* statistics used in the CAMI study, rather than relying just on similarity to results produced through alternative techniques and tools. Those organisms whose marker genes regions were not in the provided contigs have no presence in the WGS reads and have been dropped from these evaluations as they were not available for Kelpie to extract and assemble. Low abundance organisms with incomplete coverage of the marker gene region are included in the performance evaluations but noted and discussed.

## RESULTS

### EBI metagenomic pipeline

As discussed in the 'Background' section, taxonomic profiles generated by directly mapping reads to reference sets are inherently somewhat imprecise as the selected reads will be randomly drawn from the target genes (16S rRNA in this case), and will include reads covering both conserved and shared regions. In addition, these reads will be shorter than full-length amplicons, further reducing the specificity of mappings, and the accuracy of the resultant taxonomic profiles. The ERP008951 project was a faecal microbiome study that had been run using V2 of the EBI pipeline, and the selected reads had been both pair-merged and trimmed to 16S rRNA gene boundaries. Some adjustments were made to Firmicute lineages in the EBI OTU tables (e.g., moving Veillonellaceae → Negativicutes) to reduce perceived mismatches coming from the use of different taxonomies, but this taxonomic harmonisation is incomplete for lower abundance taxa.

The Kelpie-generated results are in good agreement with the EBI profile, but with fewer unclassified sequences and many more resolving to Species-level. The EBI profile shows much more diversity at higher level taxa (132 vs. 24 Orders, for example) with a long tail of rarer taxa such as *Halanaerobiales*, *Natranaerobiales* and several *Chlorobi*. Some of this diversity will be coming from the direct matching of reads from very low coverage organisms, but some will be a result of the less precise matching of short reads and matches onto conserved regions. The full results of this comparison can be found in Table S1, and Table 2 shows the 25 most abundant species (for EBI) and the 25 largest OTUs (for the Kelpie-assembled sequences). Only 5 of these top EBI classifications were resolved to species level, while all of the Kelpie sequences were assigned to a species, with the lowest

**Table 2** **Top 25 most abundant organisms found in EBI project ERP008951.** The first part of the table comes from the community profile generated by the EBI Metagenomics Portal, and the second part is from an OTU table produced from Kelpie-generated data. The highlighted cells were only resolved to a taxonomic level above Species.

**Top 25 EBI Species**

| Family | Genus | Species | Sum |
|---|---|---|---|
| Unclassified_f | Unclassified_g | Unclassified_sp | 96675 |
| Bacteroidaceae | Bacteroides | Bacteroides_sp | 242238 |
| Lachnospiraceae | Lachnospiraceae_g | Lachnospiraceae_sp | 100484 |
| Prevotellaceae | Prevotella | Prevotella copri | 86982 |
| Ruminococcaceae | Faecalibacterium | Faecalibacterium prausnitzii | 70676 |
| Ruminococcaceae | Ruminococcaceae_g | Ruminococcaceae_sp | 68736 |
| Clostridiales_f | Clostridiales_g | Clostridiales_sp | 67346 |
| Lachnospiraceae | Lachnospira | Lachnospira_sp | 38338 |
| Enterobacteriaceae | Enterobacteriaceae_g | Enterobacteriaceae_sp | 30687 |
| Bacteroidaceae | Bacteroides | Bacteroides uniformis | 24942 |
| Lachnospiraceae | Blautia | Blautia_sp | 24770 |
| Sutterellaceae | Sutterella | Sutterella_sp | 24151 |
| Porphyromonadaceae | Parabacteroides | Parabacteroides_sp | 23702 |
| Lachnospiraceae | Coprococcus | Coprococcus_sp | 21015 |
| Ruminococcaceae | Ruminococcus | Ruminococcus_sp | 20449 |
| Prevotellaceae | Prevotella | Prevotella_sp | 15063 |
| Lachnospiraceae | Roseburia | Roseburia_sp | 13965 |
| Porphyromonadaceae | Parabacteroides | Parabacteroides distasonis | 13588 |
| Rikenellaceae | Rikenellaceae_g | Rikenellaceae_sp | 13216 |
| Ruminococcaceae | Oscillospira | Oscillospira_sp | 12402 |
| Veillonellaceae | Dialister | Dialister_sp | 11469 |
| Selenomonadaceae | Megamonas | Megamonas_sp | 9059 |
| Enterobacteriaceae | Klebsiella | Klebsiella_sp | 9045 |
| Lachnospiraceae | Dorea | Dorea_sp | 8362 |
| Bacteroidaceae | Bacteroides | Bacteroides ovatus | 7631 |

**Top 25 Kelpie OTUs to Species/Accession**

| Family | Genus | Species | Match% | # == | Size |
|---|---|---|---|---|---|
| Bacteroidaceae | Bacteroides | Bacteroides dorei JCM 13471; 175 (AB242142) | 99.6 | 2 | 19769 |
| Ruminococcaceae | Faecalibacterium | Faecalibacterium prausnitzii ATCC 27768 (AJ413954) | 98.8 | 1 | 7177 |
| Prevotellaceae | Prevotella | Prevotella copri CB7 (AB064923) | 100 | 1 | 6589 |
| Eubacteriaceae | Eubacterium | Eubacterium rectale (L34627) | 100 | 1 | 4891 |
| Enterobacteriaceae | Enterobacter | Enterobacter cancerogenus LMG 2693 (Z96078) | 100 | 33 | 4706 |
| Bacteroidaceae | Bacteroides | Bacteroides finegoldii JCM 13345; 199T (AB222699) | 100 | 1 | 3719 |
| Eubacteriaceae | Eubacterium | Eubacterium eligens (L34420) | 99.6 | 1 | 2561 |
| Bacteroidaceae | Bacteroides | Bacteroides coprocola M16 (AB200224) | 100 | 1 | 2480 |
| Bacteroidaceae | Bacteroides | Bacteroides uniformis JCM 5828T (AB050110) | 100 | 1 | 2480 |
| Lachnospiraceae | Lachnospiraceae_g | Lactobacillus rogosae ATCC 27753 (NR_104836.1) | 100 | 1 | 2286 |

**Table 2** (*continued*)

**Top 25 Kelpie OTUs to Species/Accession**

| Family | Genus | Species | Match% | # == | Size |
|---|---|---|---|---|---|
| Porphyromonadaceae | Parabacteroides | Parabacteroides distasonis JCM 5825 (AB238922) | 99.2 | 1 | 1937 |
| Sutterellaceae | Sutterella | Sutterella wadsworthensis WAL 9799 (GU585669) | 100 | 1 | 1612 |
| Prevotellaceae | Prevotella | Prevotella stercorea CB35 (AB244774) | 98.4 | 1 | 1386 |
| Lachnospiraceae | Anaerostipes | Anaerostipes sp. 5 1 63FAA (JF412658) | 100 | 3 | 1336 |
| Lachnospiraceae | Blautia | Blautia luti DSM 14534 (NR_114315.1) | 100 | 2 | 1233 |
| Lachnospiraceae | Fusicatenibacter | Fusicatenibacter saccharivorans HT03-11 (AB698910) | 100 | 1 | 1174 |
| Selenomonadaceae | Megamonas | Megamonas funiformis YIT 11815 (AB300988) | 100 | 1 | 1163 |
| Porphyromonadaceae | Parabacteroides | Parabacteroides goldsteinii WAL 12034 (AY974070) | 100 | 1 | 1135 |
| Lachnospiraceae | Roseburia | Roseburia inulinivorans A2-194 (AJ270473) | 100 | 1 | 1079 |
| Bacteroidaceae | Bacteroides | Bacteroides massiliensis B84634 (AY126616) | 100 | 1 | 1063 |
| Lachnospiraceae | Clostridium XlVa | Clostridium algidixylanolyticum SPL73 (AF092549) | 97.6 | 2 | 930 |
| Lachnospiraceae | Coprococcus | Coprococcus comes ATCC 27758 (EF031542) | 100 | 1 | 922 |
| Ruminococcaceae | Gemmiger | Gemmiger formicilis ATCC 27749; X2-56 (GU562446) | 100 | 1 | 845 |
| Bifidobacteriaceae | Bifidobacterium | Bifidobacterium stercoris Eg1 (FJ611793) | 100 | 6 | 816 |
| Acidaminococcaceae | Phascolarctobacterium | Phascolarctobacterium faecium (X72865) | 100 | 1 | 808 |

match identity of 97.6%. Figure 2 shows how the EBI and Kelpie-generated taxonomic profiles compare at an Order level using charts generated by STAMP (*Parks et al., 2014*). Figure 2A is a bar chart of the 22 most abundant and statistically significant Orders across both datasets (Two-sided Fisher's exact test, Storey's FDR). Figure 2B is a scatter plot produced from the same data, with an $R^2$ value of 0.994.

## Coal seam metagenome studies

The purpose of these Coal Seam Metagenome (CSM) tests was to determine if the sequences extracted and assembled by Kelpie were comparable to, and preferably identical to, those produced using traditional amplicon sequencing, and also to demonstrate that these sequences were compatible with conventional amplicon-based pipelines and tools. The first 25 (of 228) rows from the OTU table generated from the combined amplicon and Kelpie-generated data are shown in Table 3, and the full OTU table for all three samples can be found in Table S2. The full spreadsheet underlying this table is available as Table S6 ('AE-All' tab). This OTU table shows complete agreement between the amplicon and Kelpie-generated samples until well into the rare organisms at the tail of the abundance distribution. The actual number of sequences assigned to each OTU correspond reasonably well for the two data sources, and the presence of primer bias in the amplicon PCR data means that perfect agreement should not be expected anyway.

Multivariate comparisons between the three samples analysed using the amplicon vs extended approaches were performed by permutational multivariate analysis of variance (PERMANOVA) using Primer 7 + (Plymouth Marine Laboratory, UK). Two analysis were performed, one using presence/absence data and the second using the abundance data, in both cases resemblance statistics were derived from Bray-Curtis similarities.

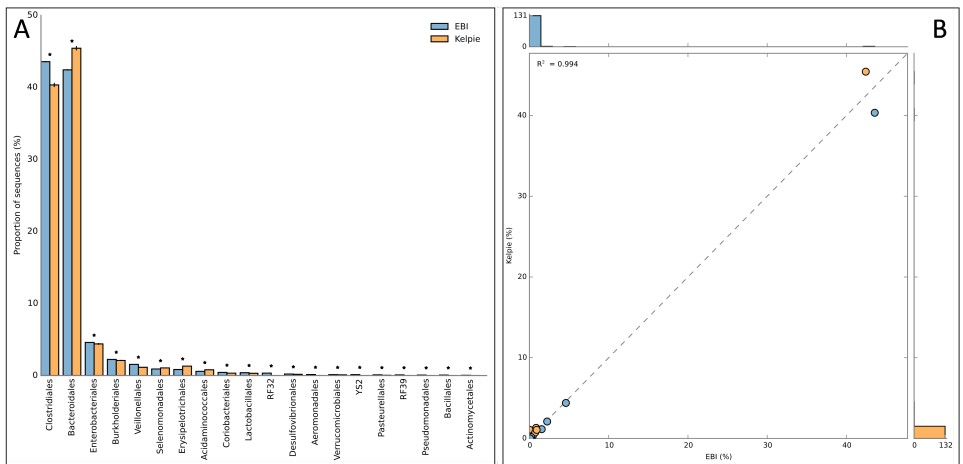

**Figure 2** **Order-level comparison between taxonomic profiles for EBI project ERP008951.** (A) Bar chart showing the most abundant Orders found by the EBI pipeline and in the Kelpie-based OTU table. (B) Scatter plot for the same data. Extracted from the spreadsheet in Table S1 and plots generated by STAMP.

Presence/absence: No difference between the two data sets were found (Pseudo-F = 1.805, $P = 0.092$). Indicating that the two-techniques produced statistically similar communities.

Abundance data: No difference between the two data sets were found (Pseudo-F = 0.233, $P = 0.702$). Indicating that the two-techniques produced statistically similar communities.

The first OTU in the combined OTU table where an amplicon does not have an equivalent Kelpie sequence occurs at a relative abundance level of 0.03%, and at a cumulative abundance of 98.8% of the amplicon sequences. Figure 3A shows how the percentage of OTUs found by using both amplicons and Kelpie varies with cumulative abundance, both for the combined OTU table and for each of the per-sample OTU tables. Figure 3B is a PCA plot produced by STAMP (*Parks et al., 2014*). STAMP ran a multiple group test using ANOVA, with a Games-Howell post-hoc test, Eta-squared effect size, and with multiple test correction done using Storey FDR. These tests showed there were just 9 'active features', and the most abundant of these (OTU_105, an unclassified *Firmicutes*) represented only 0.015% of all the amplicon reads.

There is always a considerable amount of 'noise' in amplicon-based studies, caused by effects such as PCR artefacts, cross-sample contamination in the pre-sequencing processes and 'tag-jumping' (*Dickie, 2010*; *Schnell, Bohmann & Gilbert, 2015*; *Edgar, 2016*; *Frøslev et al., 2017*), making low abundance counts somewhat unreliable. Rather than establish an arbitrary 'noise' level as a cut-off point, the read coverage for each OTU sequence from each of the 3 WGS datasets was estimated by using a kMer mapping tool, and counts for any OTU/sample cell with less than 90% kMer coverage of the corresponding sequence were not included in the performance numbers below. Some of these appear to be artefacts, and others will simply be rare organisms with insufficient WGS read coverage. These 'untrusted' amplicon counts are shown in bold in Table 3, and in red in the combined

**Table 3 Extract from CSM OTU table (amplicons and extended reads).** The first 25 of 228 rows of the Coal Seam Metagenome OTU table found in Table S2. The 'amp' columns are amplicon counts; the 'ext' columns are counts of Kelpie extended reads. Counts in bold indicate that the OTU consensus sequence was not completely covered by WGS reads.

| OTU | Size | Species | W1 amp | W1 ext | W2 amp | W2 ext | W3 amp | W3 ext |
|---|---|---|---|---|---|---|---|---|
| 1 | 43,603 | Desulfuromonas acetexigens (T) (U23140) | 27333 | 13574 | **132** | 0 | 1554 | 1010 |
| 2 | 24,970 | Thermodesulfovibrio aggregans (T) TGE-P1 (AB021302) | **24** | 0 | 17120 | 7816 | **10** | 0 |
| 3 | 10,514 | Treponema zuelzerae (T) type strain: DSM 1903; 2 (FR749929) | **13** | 0 | 1171 | 305 | 5956 | 3069 |
| 4 | 10,163 | Methanobacterium subterraneum (T) A8p, DSM 11074 (X99044) | **5** | 0 | **29** | 0 | 7736 | 2393 |
| 5 | 7,081 | Cytophaga fermentans (T) ATCC 19072 (M58766) | **9** | 0 | 5845 | 1220 | **7** | 0 |
| 7 | 6,514 | Methanosaeta harundinacea (T) 8Ac (AY817738) | 1032 | 192 | **16** | 0 | 3332 | 1942 |
| 6 | 6,264 | Parabacteroides distasonis (T) JCM 5825 (AB238922) | 1270 | 271 | **9** | 0 | 3116 | 1598 |
| 8 | 5,520 | Thermacetogenium phaeum (T) PB (AB020336) | **5** | 0 | **14** | 0 | 3285 | 2216 |
| 10 | 4,837 | candidate division OP1 clone OPB14 (AF027045) | **3** | 0 | 4057 | 771 | **6** | 0 |
| 12 | 4,611 | Lysinibacillus sp. LAM612 (KF443809) | **3** | 0 | **7** | 0 | 533 | 4068 |
| 9 | 4,258 | Methanosarcina siciliae type strain: DSM3028 (FR733698) | 1238 | 2733 | **11** | 0 | 54 | 222 |
| 13 | 3,847 | Methanocalculus pumilus (T) MHT-1 (AB008853) | 3312 | 476 | **30** | 0 | **29** | 0 |
| 11 | 3,652 | Desulfotomaculum acetoxidans (T) DSM 771 (Y11566) | **6** | 0 | 2463 | 1177 | **6** | 0 |
| 14 | 3,390 | Syntrophaceticus schinkii (T) Sp3 (EU386162) | **6** | 0 | 2871 | 506 | **7** | 0 |
| 15 | 3,383 | Methanobacterium aarhusense (T) H2-LR (AY386124) | **1** | 0 | 3104 | 271 | **7** | 0 |
| 17 | 3,012 | Methanothermobacter thermoflexus (T) IDZ, VKM B-1963, DSM 7268 (X99047) | **1** | 0 | 2685 | 326 | **0** | 0 |
| 16 | 2,920 | Sulfurospirillum alkalitolerans HTRB-L1 (GQ863490) | 2340 | 508 | **41** | 0 | **31** | 0 |
| 21 | 2,114 | Methanobacterium alcaliphilum (T) NBRC 105226 (AB496639) | **2** | 0 | 1161 | 100 | 586 | 265 |
| 18 | 2,099 | Clostridium hungatei (T) AD; ATCC 700212 (AF020429) | **5** | 0 | **5** | 0 | 1124 | 965 |
| 20 | 2,067 | Natronincola peptidivorans (T) Z-7031 (EF382661) | **12** | 0 | **8** | 0 | 1293 | 754 |
| 19 | 1,955 | Pontibacter sp. JC215 A10 (HG008901) | 4 | 0 | **2** | 0 | 931 | 1018 |
| 23 | 1,734 | Porphyromonas pogonae strain MI 10-1288 (NR 136443.1) | 1059 | 128 | **29** | 0 | 389 | 129 |
| 25 | 1,557 | Acetobacterium malicum (T) DSM 4132 (X96957) | 929 | 304 | **16** | 0 | 153 | 155 |
| 22 | 1,515 | Desulfovibrio oxamicus (T) DSM 1925 (DQ122124) | **19** | 10 | **2** | 0 | 860 | 624 |
| 24 | 1,513 | *No closest species found* | **2** | 0 | 955 | 556 | **0** | 0 |

OTU table in Table S2. This table also gives some hint about the prevalence of various artefacts in amplicon data, with dominant organisms in one sample often having very low counts in other organisms, as would be expected from the above references. The 9 'active' OTUs found by STAMP all had incomplete WGS coverage of the corresponding amplicon sequence, with the most abundant of them (OTU_105) only having 12%–30% kMer coverage.

The clustering process used when generating the OTU sequences could be masking differences between the amplicon and Kelpie-generated sequences as they only have to be

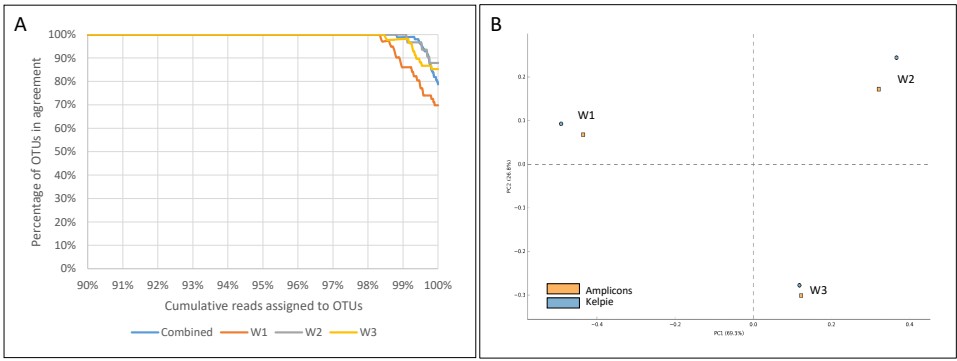

**Figure 3 Agreement between amplicon and Kelpie-based OTUs for CSM datasets.** (A) Percentages ordered by cumulative read count for the four 'AE' OTU tables in Table S6 (samples combined, and processed separately). In the combined table, the first OTU without supporting counts from both amplicons and Kelpie-extended reads comes after 98.8% of the amplicons reads have been assigned to OTUs (83rd OTU in reverse cumulative size order), and represents 0.03% of the amplicon reads. (B) PCA plot showing the similarity between the amplicon and Kelpie-based profiles.

97% similar to be included in the same OTU. To validate the actual sequences generated by Kelpie, both the Kelpie-generated 'amplicons' and the real PCR amplicons for each sample were run separately through the same amplicon pipeline, and the resulting OTU centroid sequences were compared using USearch usearch_global. The results of this sequence-level identity comparison for the W2 dataset is shown in Table 4 and the results for all 3 datasets are summarised in Table 5. The full results from these sequence-level comparisons can be found in Table S6 ('K-A' tabs).

The majority (82% to 93%) of the amplicon and Kelpie-assembled centroid OTU sequences were 100% identical, and 93% to 100% of the sequences were at least 97% identical. A close examination of these not-100% identical sequences showed that in these cases there were a number of closely related strains present in the bacterial community, and the differences between the amplicon and Kelpie-generated OTU sequences were just a result of the clustering algorithm picking a different centroid sequence from within the cluster. There are also three OTUs that are found purely in Kelpie data, with no matching amplicon reads. Aligning the WGS reads to the centroid sequences for these three OTUs indicates that they actually are present in the community, and their absence from the amplicon reads may be an artefact of biases inherent in the PCR process.

The combined OTU table from running the amplicon pipeline on the amplicons, the Kelpie-generated amplicons, the extracted 16S rRNA V4 regions from the full metaSPAdes assembly, and the metaSPAdes 16S rRNA-only V4 assembly for all three samples can be seen in full in Table S3. The counts shown in the metaSPAdes-based columns in this OTU table are derived from the stated depth of coverage for the contigs from which they were extracted. Figure 4 shows how well the two assembly-based techniques compare to the amplicon and Kelpie-generated datasets. The amplicon and Kelpie sequences both found all the most abundant OTUs (top 98%), while the two metaSPAdes assemblies missed about a third of these, including some of the most abundant ones. The full metaSPAdes

**Table 4  Details from the identity comparisons between the amplicons and Kelpie-generated OTU centroid sequences for the W2 CSM dataset.** The centroid sequences for OTUs 20 and 29 are slightly different, although within the 97% similarity threshold. Closer examination of the sequence 'clouds' that were clustered together to form these OTUs showed that these apparent differences arose as a result of the choice of different centroid sequences rather than the Kelpie and amplicon being actually different and distinct.

| OTU | Size | Kelpie species | Id% | Amplicon species |
|-----|------|----------------|-----|------------------|
| 1 | 7816 | Thermodesulfovibrio aggregans (T) TGE-P1 (AB021302) | 100 | Thermodesulfovibrio aggregans (T) TGE-P1 (AB021302) |
| 2 | 1220 | Cytophaga fermentans (T) ATCC 19072 (M58766) | 100 | Cytophaga fermentans (T) ATCC 19072 (M58766) |
| 3 | 1169 | Desulfotomaculum acetoxidans (T) DSM 771 (Y11566) | 100 | Desulfotomaculum acetoxidans (T) DSM 771 (Y11566) |
| 4 | 847 | Moorella humiferrea (T) 64 FGQ (GQ872425) | 100 | Moorella humiferrea (T) 64 FGQ (GQ872425) |
| 5 | 771 | candidate division OP1 clone OPB14 (AF027045) | 100 | candidate division OP1 clone OPB14 (AF027045) |
| 6 | 556 | – | 100 | – |
| 7 | 506 | Syntrophaceticus schinkii (T) Sp3 (EU386162) | 100 | Syntrophaceticus schinkii (T) Sp3 (EU386162) |
| 8 | 419 | Thermodesulfovibrio aggregans (T) TGE-P1 (AB021302) | 100 | Thermodesulfovibrio aggregans (T) TGE-P1 (AB021302) |
| 9 | 408 | Ignavibacterium album (T) Mat9-16 (AB478415) | 100 | Ignavibacterium album (T) Mat9-16 (AB478415) |
| 10 | 326 | Methanothermobacter thermoflexus (T) IDZ, VKM B-1963, DSM 7268 (X99047) | 100 | Methanothermobacter thermoflexus (T) IDZ, VKM B-1963, DSM 7268 (X99047) |
| 11 | 305 | Treponema zuelzerae (T) type strain: DSM 1903; 2 (FR749929) | 100 | Treponema zuelzerae (T) type strain: DSM 1903; 2 (FR749929) |
| 12 | 271 | Methanobacterium aarhusense (T) H2-LR (AY386124) | 100 | Methanobacterium aarhusense (T) H2-LR (AY386124) |
| 13 | 211 | – | 100 | – |
| 20 | 108 | Dethiobacter alkaliphilus (T) AHT 1 (EF422412) | 98 | Dethiobacter alkaliphilus (T) AHT 1 (EF422412) |
| 14 | 100 | Methanobacterium alcaliphilum (T) NBRC 105226 (AB496639) | 100 | Methanobacterium alcaliphilum (T) NBRC 105226 (AB496639) |
| 15 | 100 | – | 100 | – |
| 16 | 98 | Thermodesulfovibrio yellowstonii (T) YP87 (AB231858) | 100 | Thermodesulfovibrio yellowstonii (T) YP87 (AB231858) |
| 17 | 85 | Pelotomaculum propionicicum (T) MGP (AB154390) | 100 | Pelotomaculum propionicicum (T) MGP (AB154390) |
| 18 | 82 | Sunxiuqinia faeciviva (T) JAM-BA0302 (AB362263) | 100 | Sunxiuqinia faeciviva (T) JAM-BA0302 (AB362263) |
| 19 | 79 | Thermanaerothrix daxensis strain GNS-1 (NR 117865.1) | 100 | Thermanaerothrix daxensis strain GNS-1 (NR 117865.1) |
| 21 | 65 | Smithella propionica (T) LYP (AF126282) | 100 | Smithella propionica (T) LYP (AF126282) |
| 22 | 60 | Caldicoprobacter oshimai (T) JW/HY-331 (AB450762) | 100 | Caldicoprobacter oshimai (T) JW/HY-331 (AB450762) |
| 23 | 48 | Bellilinea caldifistulae (T) GOMI-1 (AB243672) | 100 | Bellilinea caldifistulae (T) GOMI-1 (AB243672) |
| 24 | 37 | Leptolinea tardivitalis (T) YMTK-2 (AB109438) | 100 | Leptolinea tardivitalis (T) YMTK-2 (AB109438) |
| 25 | 36 | uncultured bacterium KF-JG30-18 (AJ295656) | 100 | uncultured bacterium KF-JG30-18 (AJ295656) |
| 26 | 35 | Desulfotomaculum kuznetsovii strain 17 (NR 115129.1) | 100 | Desulfotomaculum kuznetsovii strain 17 (NR 115129.1) |
| 27 | 29 | Dethiobacter alkaliphilus (T) AHT 1 (EF422412) | 100 | Dethiobacter alkaliphilus (T) AHT 1 (EF422412) |
| 28 | 21 | Acidobacteria bacterium P105 (KJ461654) | 100 | Acidobacteria bacterium P105 (KJ461654) |
| 29 | 21 | Olegusella massiliensis strain KHD7 (NR 146815.1) | 99.6 | Olegusella massiliensis strain KHD7 (NR 146815.1) |
| 30 | 19 | Syntrophorhabdus aromaticivorans (T) UI (AB212873) | 100 | Syntrophorhabdus aromaticivorans (T) UI (AB212873) |

metagenomic assembly performs slightly worse (57% of the top 98%) than the assembly from the filtered 16S rRNA reads (65% of the top 98%). The supporting data for this chart can be found in Table S6 ('AESS' sheets).

## CAMI SYNTHETIC DATASETS

The results for the Low Complexity CAMI test dataset are summarised in Table 6. All of the extended 16S rRNA V4 sequences produced by Kelpie were classified to the exactly

**Table 5** **Summary of identity comparisons between centroid OTU sequences for the 3 CSM datasets.** The small number of not-identical species appear to be caused by the clustering algorithm choosing different consensus sequences from within a cluster of strain-level variants. There are a total of 3 OTUs that are found by Kelpie that do not appear in the amplicon data.

|  | W1 |  | W2 |  | W3 |  |
| --- | --- | --- | --- | --- | --- | --- |
| **#OTUs** | **39** |  | **30** |  | **57** |  |
| 100% identical | 36 | 92% | 28 | 93% | 47 | 82% |
| same species (97%+) | 1 | 3% | 2 | 7% | 4 | 7% |
| same genus (95%+) | 1 | 3% | 0 | 0% | 4 | 7% |
| not in amplicons | 1 | 3% | 0 | 0% | 2 | 4% |

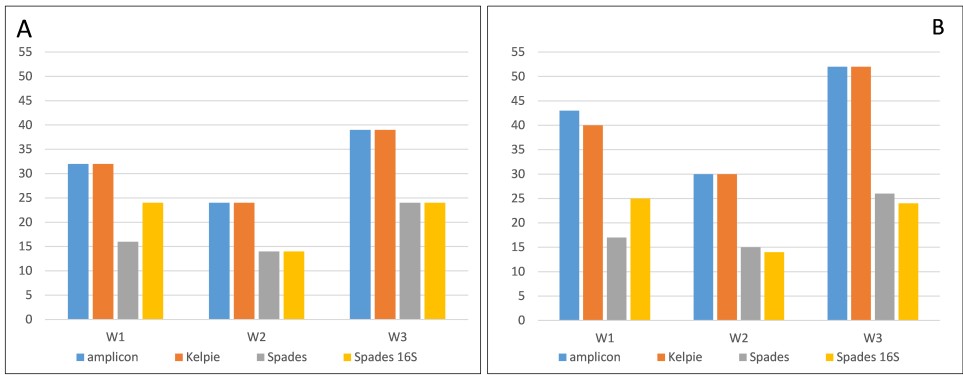

**Figure 4** **Numbers of OTUs found in the top 98% (A) and 99% (B) of the community profile for each of the three samples.** Numbers of OTUs are ranked by cumulative read count and derived from the three 'AESS-W' OTU tables in Table S6. The OTU counts have been adjusted by removing amplicon OTUs that have incomplete WGS read coverage.

same strain as the equivalent regions extracted from the assembled contigs. There are 3 cases in this test where an extracted region did not have matching Kelpie extended reads, and mapping the WGS reads back to these unmatched regions showed that they were incompletely covered in the sampled synthetic reads.

Full summaries for the tests using the CAMI Low Complexity dataset are available in Table S4, and full details in Table S7. Similarly, the comparisons for the Medium Complexity dataset are available as Table S5 (summary) and Table S8 (details). The two summary files present both the full mappings from the full CAMI 'gold' profile to the accession-matched sequences extracted from the contigs and the Kelpie-generated sequences, and the same comparisons but including just those organisms/accessions found to be present in the assembled contigs.

The CAMI paper defines two metrics that were used to assess the quality of the results submitted for their taxonomic profiling challenge. *Recall* is the percentage of organisms present in the test community were found; and *Precision* is how accurately they were identified (at various taxonomic levels). The CAMI paper gives *Recall* and *Precision* numbers for 10 profiling tools, at various taxonomic ranks, and for both the full datasets and for the top 99% of organisms by abundance. The actual results presented by CAMI
**Table 6 Comparisons for the CAMI Low Complexity dataset.** Comparison between the organisms named in the CAMI 'gold' profile, the corresponding classified rRNA V4 regions extracted from the CAMI-provided assembled contigs, and the classified Kelpie 'amplicons'. Any species in the CAMI profile whose 16S rRNA V4 region could not be found in the provided contigs has been removed from this table.

| CAMI gold profile Species | Abnd. | V4 region from contigs Species/strain | Cov% | Kelpie profile Species/strain | Reads | Abnd. |
|---|---|---|---|---|---|---|
| Schwartzia succinivorans | 28.2% | Schwartzia succinivorans strain S1-1 (NR 029325.1) | 100 | Schwartzia succinivorans strain S1-1 (NR 029325.1) | 615 | 26.3% |
| Hydrotalea sandarakina | 19.8% | Hydrotalea sandarakina strain AF-51 (NR 109380.1) | 100 | Hydrotalea sandarakina strain AF-51 (NR 109380.1) | 759 | 32.5% |
| Tetrasphaera duodecadis | 14.9% | Tetrasphaera duodecadis strain IAM 14868 (NR 040880.1) | 100 | Tetrasphaera duodecadis strain IAM 14868 (NR 040880.1) | 255 | 10.9% |
| Bacillales sp | 9.2% | Exiguobacterium acetylicum strain DSM 20416 (NR 043479.1) | 100 | Exiguobacterium acetylicum strain DSM 20416 (NR 043479.1) | 169 | 7.2% |
| Janthinobacterium sp. | 7.8% | Massilia namucuonensis strain 333-1-0411 (NR 118215.1) | 100 | Massilia namucuonensis strain 333-1-0411 (NR 118215.1) | 132 | 5.6% |
| Pseudomonas aeruginosa | 6.0% | Pseudomonas aeruginosa strain DSM 50071 (NR 117678.1) | 100 | Pseudomonas aeruginosa strain DSM 50071 (NR 117678.1) | 108 | 4.6% |
| Paracoccus denitrificans | 3.7% | Paracoccus denitrificans strain 381 (NR 026456.1) | 100 | Paracoccus denitrificans strain 381 (NR 026456.1) | 74 | 3.2% |
| Defluviimonas denitrificans | 3.0% | Defluviimonas denitrificans strain D9-3 (NR 115019.1) | 100 | Defluviimonas denitrificans strain D9-3 (NR 115019.1) | 48 | 2.1% |
| Desulfatibacillum alkenivorans | 1.9% | Desulfatibacillum alkenivorans strain PF2803 (NR 025795.1) | 100 | Desulfatibacillum alkenivorans strain PF2803 (NR 025795.1) | 42 | 1.8% |
| Actinomycetales sp. | 1.1% | Williamsia phyllosphaerae strain C7 (NR 108495.1) | 100 | Williamsia phyllosphaerae strain C7 (NR 108495.1) | 8 | 0.3% |
| Flavisolibacter ginsengisoli | 1.8% | Flavisolibacter ginsengisoli strain Gsoil 643 (NR 041500.1) | 100 | Flavisolibacter ginsengisoli strain Gsoil 643 (NR 041500.1) | 83 | 3.6% |
| Tepidibacter formicigenes | 0.7% | Tepidibacter formicigenes strain DV1184 (NR 029081.1) | 100 | Tepidibacter formicigenes strain DV1184 (NR 029081.1) | 11 | 0.5% |
| Albidovulum xiamenense | 0.4% | Albidovulum xiamenense strain YBY-7 (NR 118031.1) | 100 | Albidovulum xiamenense strain YBY-7 (NR 118031.1) | 1 | 0.0% |
| Xylella fastidiosa | 0.4% | Xylella fastidiosa strain PCE-FF (NR 041779.1) | 100 | Xylella fastidiosa strain PCE-FF (NR 041779.1) | 17 | 0.7% |
| Lampropedia hyalina | 0.4% | Lampropedia hyalina strain IAM 14890 (NR 040942.1) | 97 | *incomplete WGS coverage of region* | | |
| Lysobacter oryzae | 0.3% | Lysobacter oryzae strain YC6269 (NR 044484.1) | 100 | Lysobacter oryzae strain YC6269 (NR 044484.1) | 16 | 0.7% |
| Anaerobranca californiensis | 0.2% | Anaerobranca zavarzinii strain JW/VK-KS5Y (NR 044155.1) | 96 | *incomplete WGS coverage of region* | | |
| Nonlabens dokdonensis | 0.1% | Nonlabens dokdonensis (NR 102491.1) | 50 | *incomplete WGS coverage of region* | | |

for the various tools are not discussed further here, as there may be subtle methodological differences that would make direct comparisons difficult or unfair, especially in the treatment of strain variants. The left-hand side of Table 7 shows the *Precision* and *Recall* numbers derived from the Low Complexity dataset results, using the classified, extracted 16 rRNA V4 accessions as the 'truth' for the calculations. These metrics were only calculated at the accession/strain-level, as there were no inexact matches that needed to be resolved at higher taxonomic levels, unlike the results presented in the CAMI paper.

**Table 7    Recall and precision statistics for the CAMI Low and Medium Complexity datasets.**

|  | CAMI low complexity | | | CAMI medium complexity | | |
|---|---|---|---|---|---|---|
|  | Present in contigs | Present & fully covered by WGS reads | Top 99% by abundance | Present in contigs | Present & fully covered by WGS reads | Top 99% by abundance |
| #organisms | 18 | 15 | 14 | 71 | 51 | 57 |
| both (TP) | 15 | 15 | 14 | 51 | 51 | 49 |
| added(FP) | 0 | 0 | 0 | 0 | 0 | 0 |
| missing(FN) | 3 | 0 | 0 | 20 | 0 | 8 |
| Precision | 100% | 100% | 100% | 100% | 100% | 100% |
| Recall | 83% | 100% | 100% | 72% | 100% | 86% |

The results from the CAMI Medium Complexity dataset are similar, with the Kelpie-generated 'amplicons' always matching to the same strain identified from the extracted 16S rRNA V4 regions, and with low abundance organisms with incomplete WGS coverage having no corresponding Kelpie-generated sequences. The *Recall* and *Precision* numbers for this dataset are shown in the right-hand side of Table 7. All of these results are derived from the spreadsheets found in Table S7 (Low Complexity) and Table S8 (Medium Complexity).

Kelpie requires complete WGS read coverage of the region it is assembling, like any other assembler, and this becomes less likely with the lower abundance organisms in the community. The stochastic sampling of genomes inherent in the generation of synthetic reads will tend to produce gaps in coverage, and these coverage gaps will become more common as the simulated abundance is reduced. For the synthetic CAMI datasets, this incomplete coverage, and the subsequent decline in *Recall*, starts with organisms with about 0.4% abundance, and with an estimated depth of coverage of less than 12. The same gradual drop can be seen with the real Coal Seam WGS data, as shown previously in Fig. 3, starting there with organisms present at about 0.3% abundance in the community.

## DISCUSSION

Kelpie is a general-purpose PCR-like targeted assembler that generates sets of full-length between-primers sequences from WGS datasets. The tests and results described above all came from the same illustrative application, extracting a marker gene region for the purposes of determining community structure, but this just an example of how Kelpie can be used. This well-known application was chosen as it provided access to independently-derived community profiles and related marker gene sequences that allowed the effectiveness and accuracy of Kelpie to be compared and assessed.

The results from the coal seam metagenome study not only showed that Kelpie-generated sequences could be used to generate microbial community profiles with an accuracy and depth comparable to conventional PCR, but that the centroid sequences for the resulting Kelpie and PCR OTU clusters were either identical or found within the small 'cloud' of sequences subsumed within each cluster. These results show that Kelpie is accurately extracting at assembling between-primer region sequences from this complex WGS metagenomic data.

Both the full and 16S-filtered WGS coal seam datasets were also assembled using the metaSPAdes assembler. The results summarised in Fig. 4 show that Kelpie is more effective at handling this repeated and ubiquitous genomic region than good conventional assemblers. Kelpie extracted and assembled those distinct marker gene regions shown as present by amplicon sequencing, well down into the low depth of coverage tail, while the conventional assembler failed to produce complete sequences for many of the OTUs.

The results from the EBI and CAMI-based studies were included to show that Kelpie works equally well on WGS datasets other than coal seam metagenomes, and also that it is compatible with filtered datasets produced by conventional HMM tools. The Kelpie-derived community profiles for the synthetic Low and Medium Complexity CAMI WGS datasets closely match both the provided 'gold' profiles, and profiles built from marker gene regions extracted from the provided assembled contigs. The profiles built from both the Kelpie sequences and the extracted marker genes almost always agreed on the actual strain/accessions identified within the community, with the Kelpie data also showing some of the synthetic strains constructed from the assembled organisms. As shown in Table 7, the *Recall* and *Precision* statistics for Kelpie-generated profiles are extremely good, especially once those organisms with no coverage or incomplete coverage from the synthetic WGS reads are excluded.

Kelpie can be used for many applications other than community profiling, and just needs a pair of conserved 'primer' sequences and a filtered WGS dataset. Kelpie has recently been used to:

- Extract and assemble almost full-length 16S genes from genomic data using the 27F and 1492R primer sequences. These sequences were needed for phylogenetic analysis and prior conventional assemblies had resulted in the genes being split across multiple small contigs. This work was done in support of a study of coal seam bacteria which will be published in 2019.
- Extract and assemble bacterial 16S V4 gene regions from data produced from metagenomic 'amoebal' sequence data. These marker regions were then used to accurately classify the amoeba-associated bacteria, and allowed strain-level functional comparisons to the relevant reference organisms. This work was done as part of a study into amoebic gill disease in salmon and will be published in 2019.
- Extract and assemble multiple marker gene regions from the same WGS dataset, allowing comparisons of primer effectiveness, and improved classification accuracy (*Fuks et al., 2018*), and the use of multiple taxon-specific primers in environmental surveys.
- Extracting functional genes, such as antibiotic resistance and Nif genes from environmental metagenomic WGS datasets.

## CONCLUSIONS

The results discussed above show that Kelpie can successfully extract and assemble full length inter-primer genomic regions from whole metagenome sequencing datasets with high precision and recall, even for challenging regions such as the ubiquitous and repeated 16S rRNA gene.

Running both real bacterial 16S rRNA amplicon data and Kelpie-generated sequences through a conventional amplicon pipeline showed excellent correspondence between the 'amplicons' from both sources for all three samples until well into the low abundance tail, with the first missing OTU being found at 0.03% amplicon-based abundance. This result indicates that a Kelpie-based OTU table derived from a WGS dataset will be very close to a conventional amplicon-based table, down to the level where artefacts are starting to appear in the amplicon data. The results from the CAMI Low and Medium Complexity datasets again showed very high *precision* from the Kelpie-generated extended reads, with every extracted sequence being matched to the identical strain/accession that was assigned to the same primer-delimited regions extracted from the assembled contigs that were used as the source of the synthesised WGS reads. The *recall* shown in these dataset was also very high, up until the point was reached where the extracted regions were no longer being completely covered by the WGS reads.

The use of Kelpie in generating taxonomic profiles from WGS metagenomic reads is only an example of its potential uses, and was chosen purely because of the availability of both real and synthetic data that could be used to evaluate its effectiveness and accuracy. In practice, any region with well conserved primer sequences should be a target for extraction and assembly by Kelpie.

Apart from on-going work to improve Kelpie's accuracy when handling low abundance organisms, the only planned extension is to remove the need for the WGS data to be pre-filtered. Pre-filtering is an efficient way to reduce the size of the dataset being processed by Kelpie, allowing it to be easily kept in memory, but some target genes do not have the well-curated sets of reference sequences or HMM models that make them amenable to filtering even though they have well defined conserved regions that could be used as 'primers'. The use of pre-filtered reads is only a performance optimisation, and the first stage of Kelpie where it extracts just the small subset of WGS reads that cover the inter-primer region could be adapted to work directly from the unfiltered datasets at some performance cost.

## ACKNOWLEDGEMENTS

The DNA samples used Coal Seam study came from a commercial project funded by Origin Energy, Santos, Queensland Gas Corporation, and we thank them for permitting us to use this data in this paper. The similarity metrics for the Coal Seam metagenomes were calculated by Dr Anthony Chariton, School of Biological Sciences, Macquarie University.

### Funding

The authors received no funding for this work.

### Competing Interests

The authors declare there are no competing interests.

## Author Contributions

- Paul Greenfield conceived and designed the experiments, performed the experiments, analyzed the data, contributed reagents/materials/analysis tools, prepared figures and/or tables, authored or reviewed drafts of the paper, approved the final draft.
- Nai Tran-Dinh conceived and designed the experiments, performed the experiments, contributed reagents/materials/analysis tools, approved the final draft.
- David Midgley conceived and designed the experiments, contributed reagents/materials/ analysis tools, approved the final draft.

## Data Availability

CSIRO Data Access Portal

DOI: 10.4225/08/59f98560eba25.

DOI: 10.4225/08/5b31ca6373d48.

GitHub: https://github.com/PaulGreenfieldOz/WorkingDogs.

## Supplemental Information

Supplemental information for this article can be found online at http://dx.doi.org/10.7717/peerj.6174#supplemental-information.

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
