# Peer review of "Kelpie: generating full-length ‘amplicons’ from whole-metagenome datasets"

_PeerJ, doi:10.7717/peerj.6174_

## Round 0.1 · original submission · Minor Revisions

Both reviewers think that the work is of merit, but significant concerns have also been raised, which preclude publication of the paper in its current form. Please carefully respond to the reviewer’s comments outlined below. As pointed out by reviewer 1, there are several methodological issues that need to be clarified or better defined. These issues include 1) proper comparison to similar software applications such as EMIRGE, 2) clear discussion why or in what areas kelpie is superior to the classical and more cost-effective amplicon sequencing approach (e.g. full-length marker regions), 3) more detailed explanation of methodological problems such as retaining abundance information or thresholding low-abundance signals, as well as 4) more precise listing of the most important statistics for comparison among the tested approaches/tools. Please also improve the repository by providing detailed installation instructions, example data and walkthrough, as well as the relevant license. As pointed out by reviewer 2, I also think that readability of the manuscript can be significantly improved by streamlining the structure, avoid redundancies, and provide proper referencing. Overall, I would appreciate to see a clear statement of the pros and cons of kelpie versus the classical amplicon approach.

·

Basic reporting

This paper describes a piece of software which solves a worthwhile problem, gene-targeted assembly from short reads in the metagenomic context. Provided it works as well as the authors think it does, and that it is supported/maintained, it may facilitate lower-cost targeted-gene analysis in large datasets by limiting the scope to a handful of molecular targets.

This paper should cite and offer a one-sentence comparison to EMirge (Miller https://doi.org/10.1186/gb-2011-12-5-r44) , which does something similar.

Excessive conservation not the only cause of assembly failure; the inverse problem--insufficient coverage--is also severe but not fixable. You handle this by simple thresholding (like everyone does) but the crude effects of low-abundance signals are not described clearly. I can't tell reading the text how completely the kelipe assemblies appear to represent the short reads. I didn't catch how many different kelpie amplicons were produced for each of the datasets, and how many short reads mapped to them without digging into excel tables.
Can you mention the amplicon/shotgun quantity-quality tradeoff?

"that rarer organisms in the community will not be represented in the set of extended reads if their coverage is incomplete"
This is an important point, both to clearly define "incomplete coverage" -- here you seem to mean coverage gaps that are impossible to asseble across--and to articulate this as a limitation of the technique.

"generally good agreement" -- how good? You should find a statstic suitable for evaluating the concordance of multinomial samples for comparing kelpie to amplicons and gold standard. Euclidean distance on the p's or the multinomial generalization of Jaccard index, (called Ružička index) would both serve. Either would give a number between 0 and 1 for how similar the distributions are that you should report for all the relevant comparisons (and in the text).

"...is more flexible than amplicon sequencing" ??? This isn't really what you mean. It's more expensive than amplicon sequencing.. it targets almost all of the genes, at a sequencing depth cost, but this isn't flexibility, it's a tradeoff.

"down to very low WGS coverage depths" -- Statements like this require reference ranges. How low is very low? How do depths constructed using kelpie from WGS compare to amplicon?

"only a single viable next kmer" implies that the approach is sensitive to [sequencing] error rate. Is there a model for this, or an SOP that calls for kmer-based error correction?

Experimental design

I don't think the validation convincingly shows equivalence to amplicon analysis, and the visualizations are not inspiring, but as the authors point out, convincing validation in the wild is exceedingly difficult.

The comparisons that focus on the difference between 0 and nonzero overlook the fact that every taxon has a number from both datasets. The .03% amplicon abundance, for instance, is less relevant than the corresponding shortread mapping abundance on the SSU, which is difficult to estimate from the text.

There is a lengthy discussion in lines 410-435 about data cleaning that seems out of place. The CAMI data has some low-abundance things, and you know their abundance from the gold standard, so identifying the low-abundance taxa by mapping seems unecessary. It would be more helpful to describe the abundance levels that keplie retrieved SSUs at and the abundance levels where it failed to assemble--identifying the sensitivity boundary for keplie using the variation in CAMI abundances--rather than describing the CAMI dataset by mapping. I hope this isn't to justify suppressing the low-abundant taxa from the comparison.

Table 6 seems to evaluate presence-absence of taxa, which overlooks the numbers of reads mapped. The numbers matter, and the taxa not detected shouldn't matter much for the reason they weren't detected.

"only a single viable next kmer" implies that the approach is sensitive to [sequencing] error rate. Is there a model for this, or an SOP that calls for kmer-based error correction?

Validity of the findings

How do the results of keplie amplicon assembly retain abundance information? Are there any reads that end up unmapped to kelpie assemblies? You probably wrote it but I missed it.

Source:
The repository linked does not contain how-to style installation instructions.
The repository does not contain sample data or a walkthrough invokation on sample data. This is extraordinarily helpful for scientific programs, and will increase the audience of people who will be willing to use the software.

License:
The repository does not contain a copy of the relevant MIT license.

"down to very low WGS coverage depths" -- Statements like this require reference ranges. How low is very low? How do depths constructed using kelpie from WGS compare to amplicon?

Additional comments

Most generally, I would like to see more numbers in the text (as opposed to the figures) when the numbers are critical to the results.

The text inspires the question "Why does keplie work better for repeated regions than other assemblers?" You might speculate about what sets kelpie-assembly apart.

inherent presence --> presence (too many uses of inherent this paragraph)

Stylistic complaints: Does kmer really need camel case styling if it's not in a product name? Do amplicons really require "scare quotes" unless you are referring to virtual amplicons, in which case, isn't adding an epithet more clear than punctuation ? Similarly 'gold' taxonomic profiles can be gold-standard taxonomic profiles.

The abbreviation "CSM" for Coal Seam Microbiome used in the captions to the figures should be spelled out in the text.

The names of underlying in-house tools (FilterReads) need not be buried in footnotes.

·

Basic reporting

1) Literature references, sufficient field background/context provided

In general I found the introduction clear but a bit poor of reference. As example:

Sentence at line 53-54 need more support, please provide some reference to support it or improve the description of the phenomena.

Sentence at line 63-64 need more support, please provide some reference to support it.

The sentence at line 67 is a bit redundant with the previous one (Line 53-54) and need to be supported by references.

The paragraph from line 75 to line 84 is a good summary of amplicon method but need to be improved with some references.

2) Professional article structure, figs, tables. Raw data shared.

In general the structuring of some paragraphs should be revised to make the reading more fluid and the results more comprehensive.

Some issues should be stressed more in the introduction to highlight the results. For example, the problem of PCR artifacts have a key role in environmental molecular studies but this problem is introduced in the results section, passing into the background (Line 334-336)

The section Results contains several paragraphs that I think should be moved in "Materials and Methods" section.
The section "Materials" is not present and the description of the database are in the result section. This type of structure makes the reading more cumbersome.

More in details:
Line 276-295, Line 304-312 both paragraphs sounds more as “materials”. For the sake of clarity I suggest to move in this section the descriptions of the datasets used for the benchmarks.

Line 319-323 This description should be moved in Methods
Line 369-373 This description should be moved in Methods

Experimental design

I advise the authors to carefully review the materials and methods section and structure it more clearly as suggested in the previous box, maybe using a drawn workflow.

Validity of the findings

no comment

Additional comments

Kelpie seems to be a very interesting tool, with several possibilities of applications not only for the 16S (as suggested by the authors but unfortunately not tested or not showed). A more linear text structure could make reading more usable even by readers outside the bioinformatics field.

---

## Round 0.2 · accepted · Accept

Dear Paul

I am pleased to inform you that your paper has been accepted for publication. I appreciate your thorough revisions and hope your software application will have an impact on the field.

Sincerely,
Martin Hartmann

# ·

Basic reporting

no comment

Experimental design

no comment

Validity of the findings

no comment

Additional comments

I thank the authors for the remarkable improvement in the linearity of the text. The parts added or repositioned made the reading smoother. This clarified the potential of the software and highlighted both the positive sides and the possible limits, making the reader more aware of the instrument he is using.
I still say that even if it does not bring upsetting news in the field of metagenomics (and without having pre-existing ones) it is a simple and well-made tool that fulfills the one for which it was written.
Honestly, I've never seen it as an alternative to the amplicon base approach but more like a parallel approach to fill some gaps. In the ecological field (i.e. metabarcoding) it can be an interesting tool to test, especially in cases where the length of the marker is a limit in resolving the taxonomy.
In my opinion the work is certainly worthy of publication.